# Gut Microbiome Changes in Gestational Diabetes

**DOI:** 10.3390/ijms232112839

**Published:** 2022-10-25

**Authors:** Ruxandra Florentina Ionescu, Robert Mihai Enache, Sanda Maria Cretoiu, Bogdan Severus Gaspar

**Affiliations:** 1Department of Cardiology I, Central Military Emergency Hospital “Dr Carol Davila”, 030167 Bucharest, Romania; 2Department of Morphological Sciences, Cell and Molecular Biology and Histology, Carol Davila University of Medicine and Pharmacy, 050474 Bucharest, Romania; 3Department of Radiology and Medical Imaging, Fundeni Clinical Institute, 022328 Bucharest, Romania; 4Surgery Department, Carol Davila University of Medicine and Pharmacy, 050474 Bucharest, Romania; 5Surgery Clinic, Bucharest Emergency Clinical Hospital, 014461 Bucharest, Romania

**Keywords:** gut microbiome, microbiota, gestational diabetes, dysbiosis, *Firmicutes/Bacteroidetes* ratio, prebiotics, probiotics

## Abstract

Gestational diabetes mellitus (GDM), one of the most common endocrine pathologies during pregnancy, is defined as any degree of glucose intolerance with onset or first discovery in the perinatal period. Physiological changes that occur in pregnant women can lead to inflammation, which promotes insulin resistance. In the general context of worldwide increasing obesity in young females of reproductive age, GDM follows the same ascending trend. Changes in the intestinal microbiome play a decisive role in obesity and the development of insulin resistance and chronic inflammation, especially in patients with type 2 diabetes mellitus (T2D). To date, various studies have also associated intestinal dysbiosis with metabolic changes in women with GDM. Although host metabolism in women with GDM has not been fully elucidated, it is of particular importance to analyze the available data and to discuss the actual knowledge regarding microbiome changes with potential impact on the health of pregnant women and newborns. We analyzed peer-reviewed journal articles available in online databases in order to summarize the most recent findings regarding how variations in diet and metabolic status of GDM patients can contribute to alteration of the gut microbiome, in the same way that changes of the gut microbiota can lead to GDM. The most frequently observed alteration in the microbiome of patients with GDM was either an increase of the *Firmicutes* phylum, respectively, or a decrease of the *Bacteroidetes* and *Actinobacteria* phyla. Gut dysbiosis was still present postpartum and can impact the development of the newborn, as shown in several studies. In the evolution of GDM, probiotic supplementation and regular physical activity have the strongest evidence of proper blood glucose control, favoring fetal development and a healthy outcome for the postpartum period. The current review aims to summarize and discuss the most recent findings regarding the correlation between GDM and dysbiosis, and current and future methods for prevention and treatment (lifestyle changes, pre- and probiotics administration). To conclude, by highlighting the role of the gut microbiota, one can change perspectives about the development and progression of GDM and open up new avenues for the development of innovative therapeutic targets in this disease.

## 1. Introduction

Gestational diabetes mellitus (GDM) represents a subcategory of diabetes mellitus, being defined as any degree of glucose intolerance, with debut/first discovery during pregnancy [1,2,3]. GDM is one of the most frequent endocrine pathologies in pregnant women [4], affecting about 5–15% of pregnancies worldwide, although an important role regarding its prevalence is played by ethnicity [3,5,6]. GDM can be determined by a multitude of risk factors, a significant role being played by dietary intake and body mass index (BMI). These have to be modulated by lifestyle interventions to assure a normal microbiota during pregnancy, in order to improve the health of both the mother with GDM, as well as the child. The most important risk factors, identified by the majority of studies, are summarized in Table 1.

Insulin resistance during pregnancy is caused by the action of placental hormones, i.e., human placental lactogen and placental growth hormone [14]. In a normal pregnancy, there is an increase in metabolic demand and insulin resistance, and therefore, the number of maternal β-cells increase by hypertrophy and neogenesis, accompanied by insulin production and secretion [15]. In women with high insulin resistance, the pancreatic β-cells fail to adapt accordingly to the intensifying metabolic needs, and glucose intolerance and hyperglycemic status appear [16]. The evolution of glycemic status in normal pregnancy and in a GDM pregnancy is illustrated in Figure 1.

In the last few years, studies have shown the importance of metabolic status changes in GDM and its effects on altering the gut microbiome, but also the fact that disturbances of gut microbiota can be a contribution in developing GDM [18,19]. During normal pregnancy, physiological changes in hormonal, metabolic, and immunological homeostasis can lead to an intense inflammatory process, while insulin resistance is increasing, in association with proliferation and alteration of bacterial load in the gut [18]. This can exert influence on intestinal permeability and energy balance, inducing some of the pathogenesis of GDM [20]. The exact correlation between these processes remains unknown [21]. Additionally, GDM can have predictive values for obesity among children and also for the development of inflammatory diseases in offspring [22]. As the gut microbiome is influenced by our diet or immune system, it has been shown that fecal samples differ in microbiota composition between healthy women and those with GDM, and also within the groups of women with GDM, from the first trimester to the third one [23].

Unfortunately, there are just a few studies that have analyzed the changes in the microbiome of the patients with GDM and the comparisons are difficult to make due to distinctive variations of cohorts. However, most of the results have shown either an increase of *Firmicutes* and a decrease of *Bacteroides* and *Actinobacteria* or no difference in the microbiota of the patients with GDM compared to healthy patients [8,24,25,26]. An important fact is that those changes can remain postpartum or can influence the fetus, so one of the primary objectives of the lifestyle interventions has to be the adjustment of microbiota during the perinatal period, as studies have reported that this improves the health of GDM women and of their offspring [8,20,27].

In this review, we set out to analyze the effects of changes in the intestinal microbiome in triggering GDM, the impact of dysbiotic changes on the immune system, to which it is directly related and which is also incriminated in the occurrence of GDM. Moreover, we evaluated the value of diet and lifestyle changes, as well as the effect of the administration of pre- and probiotics in the regulation of dysbiosis and the evolution of GDM. A better understanding of the interrelations between microbiota, immune system, and GDM will allow in the future to create different strategies for a therapeutic approach in preventing GDM installment and evolution.

## 2. Gut Microbiota Evolution during Normal Pregnancy

Microbiota evolves along with us from birth throughout life. Recent studies refuted the fact that microbiota develops after delivery and confirmed the existence of the microbial communities in amniotic fluid, the womb, placenta, and meconium [28,29]. During pregnancy, physiological changes that aim to maintain the health of the mother and the fetus, can modify the microbiota of the gut, oral cavity, and vagina of the mother [30]. First of all, the gut microbiota presents significant modifications during the course of a normal pregnancy, from increased α-diversity (within individuals) and decreased β-diversity (between individuals) in the first trimester, to a reversed α-β ratio in the third trimester. All of these changes are substantially influenced by a lot of factors, such as diet, body mass index, GDM, antibiotics, endocrine, and immune systems of the host [18,31].

Gut microbiota consists of five major bacterial phyla: *Firmicutes*, *Bacteroidetes*, *Actinobacteria*, *Proteobacteria*, and *Verrucomicrobia* [32]. The *Firmicutes* phylum is made up of *Ruminococcus*, *Clostridium*, *Lactobacillus*, and butyrate-producing bacteria, and the *Bacteroidetes* phylum is represented by *Bacteroides*, *Prevotella*, and *Xylanibacter* [32,33]. *Bacteroidetes* and *Firmicutes* represent the main components that can be correlated with obesity and type 2 diabetes (T2D).

During pregnancy, in all the body’s microbial communities, there are specific changes. One can detect an increase in oral concentrations of *Porphyromonas gingivalis*, *Aggregatibacter actinomycetemcomitans*, and *Candida*. At subgingival plaque sites, populations of *Porphyromonas gingivalis* and *Aggregatibacter actinomycetemcomitans* were found to be higher in the beginning and middle period of the pregnancy than in non-pregnant women [34,35]. Other studies suggest that, in the second and third trimester, there are elevated concentrations of subgingival *Aggregatibacter actinomycetemcomitans* and periodontal *Candida* [34,35,36,37]. It was underlined that progesterone and estrogen contribute to the microbiota changes, but the exact mechanisms remain unclear, except for the known fact that estrogens can accentuate *Candida* infections [34,35,36,37].

In the placenta, aerobic and anaerobic bacteria are found in higher quantities. In the gastrointestinal tract, *Actinobacteria* and *Proteobacteria* multiply, while *Faecalibacterium* levels decrease. In the last trimester, there are significant populations of *Streptococcus*, *Lactobacillus*, and *Enterococcus*. Immediately postpartum, *Streptococcus* populations are still substantial, while the density of *Faecalibacterium*, butyrate-producing bacteria also possessing anti-inflammatory roles, is decreasing [18,38]. Regarding the vaginal microbiota, *Lactobacillus* species develop throughout the pregnancy with regression after birth [18,35].

Throughout the first trimester, studies have shown that the gut microbiota of a pregnant woman consists mostly of *Firmicutes*, principally *Clostridiales*, over *Bacteroidetes*, and mucin-degrading microorganisms like *Akkermansia* (*Verrucomicrobia* phylum) and *Bifidobacterium* (*Actinobacteria* phylum) [39,40]. The presence of an increased amount of mucin-degrading microorganisms in the first trimester, that seems to remain elevated during the time of gestation, is extremely important because it increases the energy extraction (even in the absence of satisfactory nutritional substrate). Any alteration of mucin, which is necessary for the preservation of the mucosal barrier of the gut, may breach its integrity [41]. *Clostridiales*, butyrate-producing bacteria, was found in low concentrations in obese patients, T2D patients, and GDM in the second trimester [24,42,43,44,45]. The *Enterobacteriaceae* family was correlated with HbA1C levels in previous studies, being more abundant in the second trimester in GDM women and in type 2 diabetic patients compared to control groups [24,46,47].

During the course of the pregnancy, in the third trimester, proinflammatory microorganisms from *Proteobacteria* and *Actinobacteria* phyla are increased and anti-inflammatory microorganisms from *Faecalibacterium* genus (*Firmicutes* phylum, *Clostridiales* order) are decreased. As a consequence, all of these proinflammatory microorganisms can cause inflammation-related dysbiosis and changes similar to metabolic syndrome. Furthermore, obesity, hyperglycemia, and insulin resistance can appear, in association with the possibility of developing inflammatory bowel diseases and also respiratory diseases [48,49,50]. A very interesting fact is that, even if all of these microbiota changes can influence maternal health during pregnancy, they are mandatory for a normal fetal growth, as increased maternal energy storage is essential for lactation and gluconeogenesis during the neonatal period [18,51]. The most important changes of the microbiota during pregnancy are summarized in Figure 2.

Maternal microbiota during pregnancy can modulate the gut microbiota of the new-born, a few studies showing increased *Lactobacillus*, *Clostridiales*, *Bacteroidales*, and *Actinomycetales* in the vagina, microorganisms acquired during vaginal birth, protecting the newborn from atopic diseases and even from some pathogens [52,53].

Due to the physiological changes pregnant women go through, they are likely to gain weight, with BMI being another important factor that can influence the gut microbiota. The *Bacteroidetes* phylum is usually decreased (50% reduction), while the *Firmicutes* phylum is increasing (compensatory 50% increase), as suggested by a study performed on microbiota of genetically obese *ob/ob* mice, lean *ob/+* and wild-type siblings and their *ob/+* mothers [54].

We have to emphasize the importance of maternal factors that modulate the microbiota during a normal pregnancy. For example, one of the most important factors and the one we can easily modify for a healthy outcome is the dietary intake. Studies have shown that a diet based on meat can increase the *Firmicutes* phylum, while vegetarian diets reduce *Proteobacteria*, *Bacteroidetes*, and *Actinobacteria* phyla [55,56].

Diets high in fat also decrease the *Actinobacteria* phylum (responsible for inhibiting some pathogens and producing essential vitamins), while the ones based on fibers can lower the ratio of *Firmicutes* to *Bacteroidetes*, protecting from excess adiposity and gaining weight [57,58,59].

Dietary influences in gut microbiota composition are summarized in Table 2.

Unfortunately, from all of these complementary factors that can modulate the gut microbiota during normal pregnancy, dietary intake and BMI are only two from a multitude. It is difficult to understand all the interrelated mechanisms and their effects on increasing/decreasing the abundance of the microorganisms from the first to the third trimester. All the known risk factors have positive/negative consequences over the maternal gut microbiota, and further research is needed in order to obtain a better result in maintaining the health of both the mother and the child.

## 3. Dysbiotic Changes in Pregnant Women Developing Gestational Diabetes

Animal studies revealed that mice with obesity induced by nutrition displayed more abundant *Firmicutes* populations when compared to mice with normal weight [69]. A high *Firmicutes/Bacteroidetes* ratio was also found in mice fed with high-fat food compared to lean mice [61].

When comparing T2D patients to non-diabetic patients, lower quantities of *Firmicutes* phylum and *Clostridia* class were found, along with higher proportions of *Bacteroidetes* and *Proteobacteria*. As a conclusion, *Firmicutes* to *Bacteroidetes* ratios were significantly and positively correlated with reduced glucose tolerance [70].

GDM women have a low abundance of intestinal microbiota, which is associated with a proinflammatory status and insulin resistance [71]. When compared to normoglycemic pregnant women, GDM patients had elevated concentrations of *Faecalibacterium* and *Anaerotruncus* and lower concentrations of *Clostridium* and *Veillonella* [72]. *Bacteriodes* and *Isobaculum* were found to be in low concentrations in patients with GDM in the last trimester and the postpartum period [72].

Crusell et al. concluded that there were higher concentrations of *Actinobacteria phylum and Collinsella*, *Rothia*, and *Desulfovibrio* genera in GDM patients diagnosed in the third trimester. Furthermore, alterations of the gut microbiota were still present even after 8 months after birth. The abnormal composition of gut microbiota of women with GDM seemed to be similar to non-pregnant patients with T2D [72].

GDM can also have an impact on the intestinal homeostasis of the newborn. GDM, along with high values of BMI can modify gut microbial structure, variety, and short chain fatty acids (SCFA) concentrations in neonates. Compared to a healthy population, lower levels of *Lactobacillus*, *Flavonifractor*, *Erysipelotrichaceae*, and unspecified families from the *Gammaproteobacteria* were found in infants with GDM mothers. This condition was also linked to the presence of high concentrations of microorganisms responsible for early immune cell function suppression, i.e., *Phascolarctobacterium* [22].

GDM was linked to the microbiota of newborns from GDM mothers. Low levels of *Lactobacillaceae* can have a negative impact on early immunological development, as this taxon was indicated to be involved in the innate immune system evolution at young ages [22]. Newborns from mothers with GDM possessed a significant association with high *Lachnospiraceae* concentrations, which are known to be in elevated amounts in patients with GDM [21,72,73] and T2D.

Aiming to explore microbial biomarkers for GDM, Ma Shujuan and colleagues conducted a case-control study based on an early pregnancy follow-up cohort. Considerable differences were noted regarding concentrations of several microorganisms. In the GDM group, *Eisenbergiella*, *Tyzzerella 4*, and *Lachnospiraceae* NK4A136 were more abundant, while *Parabacteroides*, *Megasphaera*, and *Eubacterium eligens* group prevailed in the control group. Fasting blood glucose concentrations were positively associated with higher concentrations of *Eisenbergiella* and *Tyzzerella* 4, while three genera from the control group displayed the contradictory phenomenon (*Parabacteroides*, *Parasutterella*, *Ruminococcaceae* UCG 002) [73]. The conclusion of the study by Ma and colleagues was that dysbiosis in early pregnancy was correlated with the occurrence of GDM and that microbiota-targeted biomarkers could represent possible predictors of GDM [73].

Dissimilarities between GDM and normoglycemic pregnant women regarding intestinal microorganisms were noted in many studies [21,24,72,74,75,76,77]. In GDM women, opposed to normoglycemic controls, several differences were reported: abundant populations of *Klebsiella variicola*, *Ruminococcus*, *Prevotella*, *Desulfovibrio*, *Rothia*, *Fusobacterium*, *Blautia*, *Eubacterium hallii* group, and decreased populations of *Bifidobacterium spp.*, *Eubacterium spp.*, *Bacteroides*, *Parabacteroides*, *Dialister*, *Akkermansia*, *Marvinbryantia*, *Anaerosporobacter*, and *Faecalibacterium* [24,72,76,77,78].

## 4. Immune-Mediated Reactions and Diabetes

The pathogenesis of both type 1 diabetes (T1D) and T2D, along with all other intermediate forms of diabetes, concerns the immune system. Both inflammation and autoimmunity play roles in the development of these diseases. Changes in everyday lifestyle and diet patterns are reflected in alterations of the gut microbiota. The microbiome displays an essential role in the training and development of both the innate and adaptive immune system, while the immune system influences the symbiosis between the host and its microorganisms [79].

It is believed that immune-mediated reactions, which are induced by changes in the microbiota composition, can be facilitators for the development of diabetes, in patients with predisposition to the disease [33]. The potential causality between commensal microbiota and host immunity was documented through germ-free animal models. The lack of microbes was linked to serious gut abnormalities regarding the lymphoid tissue architecture and thus its immune roles [80].

Chronic inflammation can lead to various metabolic disorders, including atherosclerosis, obesity, and even diabetes mellitus. The interactions between immune and parenchymal cells in very active organs with metabolic roles play important parts in the development of metabolic disorders [81]. Additionally, metabolites from the gut microbiome can pass through the gut barrier and go into the systemic circulation, facilitating metabolic inflammation [82].

Communications between the immune system and the gut microbiota can also be involved in T1D [83]. Myeloid differentiation primary response 88 protein (MyD88) represents a connector for a multitude of innate immunity receptors responsible for the signaling pathways triggered by effector molecules interleukin-1 (IL-1) and interleukin-18 (IL-18) [84]. In animal studies, diabetic mice, but not obese, without MyD88 signaling developed T1D, while the existence of microbes can diminish the development of the disease. Furthermore, MyD88 regulates the differentiation of T cells and supports the homeostasis of the microbiota by acting on IgA. MyD88 also controls the development of Th17 cells through diminishing the growth of segmented filamentous bacteria in mice [85]. Reduction or absence of *Akkermansia muciniphila* populations can lead to systemic translocation of endotoxin-activated CCR+ monocytes, which can trigger the innate pancreatic beta 1a-cells, leading to high insulin resistance [86]. 

Gut microbiome and the immunological response of the host are strongly intertwined. Gut dysbiosis can contribute to immune disorders, such as inflammatory bowel disease and lupus erythematosus [87,88,89,90]. It was noted that a loss of butyrate-producing bacteria was associated with gut inflammation, with elevated levels of IL 17 and low levels of IL 10 [91]. Butyrate, in early pregnancy, can increase the chances of embryo survival in animal studies [89,92]. High local concentrations of IL-15 are cited to be associated with unfavorable pregnancy outcomes [11,89]. Elevated expression of IL-15 in the gut epithelium can change the microbiota, lowering butyrate-producing bacteria and therefore butyrate levels [89,93]. In vitro research shows that progesterone can lower the bacterial load, but increases the growth of *Faecalibacterium*, *Bacteroides*, and *Bifidobacterium* [89,94]. Estrogen was associated with increased mucosal expression of regulatory B-cells and M2 macrophages, thus improving the local barrier, with increasing concentrations of bacteria with immune modulatory roles. Additionally, both progesterone and estrogen can enforce the barrier capacity of the epithelium, which can further influence the bacterial composition [89,95].

Another mechanism that has been shown to increase the risk of autoimmune diabetes is the alteration of pancreatic β-cells function by some bacterial toxins. According to Mayers et al., who evaluated the connection between dietary microbial toxins and type 1 diabetes, they concluded that Streptomyces toxins and bafilomycin A1 can affect islet homeostasis by releasing autoantigens. In the case of a genetically predisposed person, this could lead to type 1 diabetes [96].

As many studies have shown, physical exercises can regulate the gut microbiota in T1D. Codella et al. presented the importance of training impact on the inflammatory status and glycemic profiles, and showed that in non-obese diabetic mice, exercise of moderate intensity can lower glucose effects in later stages of diabetes, pointing out that the impact of training in immunomodulation can lead to improved treatments in the future, therefore further studies are needed [97].

In cases of T2D, obesity, being characterized by a low-grade inflammation, represents one of the most important factors that can induce activation of the immune system. Gram-negative bacteria of the gut contain lipopolysaccharides (LPS) in their outer membrane that can induce an important inflammatory response [98]. By overpassing the intestinal barrier, LPS reach the systemic circulation and stimulate the receptor protein CD14, with the help of LPS-binding protein, forming a complex that can bind macrophages’ Toll-like receptor 4 and activates the production of inflammatory effectors, like activator protein 1 and nuclear factor kB [99,100]. Moreover, Pomie et al. evaluated intestinal retinoic-acid-receptor-related orphan nuclear receptor gamma (RORγt)-generated Th17 cells in T2D and showed that modification of the balance between T helper 17/regulatory T cell can induce reduction of RORγt+ and IL-17-producing CD4+ T-cells that determine insulin resistance [101].

Th1/Th2 lymphoid cell balance is also involved in the pathogenesis of diabetes mediated by gut microbiota. Some studies showed that Gram-negative and Gram-positive bacteria can trigger Th1 synthesis, by stimulating inflammatory cytokines or interferon gamma (IFN-γ) [102,103]. Furthermore, Th1 secretes IL-12 that binds some specific receptors on pancreatic β-cells and induces apoptosis and complications of T2D [104]. Ali et al. evaluated the importance of IL-12 in angiogenesis in T2D and concluded that a deficiency of IL-12 has beneficial effects on angiogenesis, with reduction of inflammation and oxidative stress [105].

Thus, in the presence of predisposing conditions like genetics for T1D and obesity for T2D, gut microbiota plays an important role by modulating adaptive and innate immunity leading to diabetes [33].

It was supposed that placental and visceral adipose tissues influence low-grade inflammation, which involves abnormal infiltration, differentiation, and activation of maternal innate and adaptive immune cells [106].

High blood glucose levels in GDM patients are linked to high placental inflammation [107]. High glucose concentrations can stimulate inflammasome activation in trophoblasts, leading to secretion of inflammatory cytokines (such as IL-1β, IL-6, IL-8, GRO-α-growth regulated oncogene alpha, RANTES-regulated on activation, normal T cell expressed and secreted, and G-CSF-granulocyte-colony stimulating factor), and antiangiogenic factors, like sFlt-1 (soluble fms-like tyrosine kinase) and sEndoglin. Hyperglycemia can also decrease trophoblast migration. The inflammatory effects were partially reduced by metformin, according to Han and colleagues [106,108].

GDM is associated with high postprandial free fatty acids concentrations and insulin resistance. The adipose tissue of the mother is also involved in the development of GDM. It is involved in the production of free fatty acids. Their concentrations rise as the pregnancy progresses, along with suppression of lipolysis and serious insulin resistance.

Concentrations of adipose tissue insulin receptor substrate (IRS)-1 protein, lipoprotein lipase and fatty acid-binding protein-2 mRNA were lower, while p85alpha subunit of phosphatidylinositol 3-kinase levels were double in the group of GDM patients compared to obese pregnant women without GDM. Low IRS-1 may play a part in insulin suppression of lipolysis as the pregnancy evolves [106,109].

In the pathophysiology of GDM are involved multiple immune cells: high NK cell cytotoxicity, monocyte activation, accentuated Th1 and Th17 response, high cytotoxic T cell, B cell and platelet count and activation, excessive infiltration and neutrophil overactivation, high insulin resistance, macrophage infiltration and activation [106].

## 5. Probiotics and Dietary/Lifestyle Changes—The Value in the Evolution of Gestational Diabetes. Prevention and Treatment

### 5.1. Prevention

Probiotics are known to efficiently control the structure and roles of the gut microbiome, diminishing adverse metabolic outcomes produced by pathogen microorganisms [110,111,112,113]. Whether they are effective as an intervention in GDM still remains an intriguing subject for debate [110].

Probiotics administered during pregnancy revealed better pregnancy and metabolic results for the mother [110,114,115]. Their beneficial effects may be due to amplifying insulin sensitivity, enzymatic deconjugation of bile acids, and transformation of cholesterol into coprostanol at intestinal level [110,116,117,118,119,120].

Taylor and colleagues aimed to review the effects of probiotic administration on fasting plasma glucose, insulin resistance, and LDL-cholesterol levels in GDM patients. However, after analyzing four high-quality randomized controlled trials, the results were that probiotic supplementation was not efficient in lowering fasting plasma glucose and LDL-cholesterol in GDM patients, but it decreased insulin resistance after a period of 6–8 weeks of probiotic supplementation [121]. Probiotic administration for a period of 4 weeks during pregnancy did not influence maternal fasting plasma glucose, according to the work of Lindsay et al. [122].

The study conducted by Luoto Raakel and colleagues aimed to identify the safety and efficacy of perinatal probiotic supplementation, for 24 months after birth, in a group of 256 women, who were randomized in their first trimester of pregnancy into a control or dietary change group [123]. The intervention category was later double-blindly randomized into placebo (diet/placebo) and probiotics supplements (diet/probiotics: *Lactobacillus rhamnosus GG* and *Bifidobacterium lactis BB-12*). The incidence of GDM was lower in the diet/probiotics group, also with safe outcomes of the pregnancy for both the mother and the child [123,124,125].

The utility of probiotics, used as a preventive method for GDM, is controversial. The efficacy of probiotic ingestion in the treatment of overweight and obese women with an oral probiotic was not effective in preventing GDM. The SPRING (the Study of Probiotics IN the prevention of Gestational diabetes) study is a multi-center, prospective, double-blind randomized controlled trial conducted in Brisbane, Australia [126]. This trial included women with BMI over 25.0 kg/m2, followed for a period of two years, and consisted of administering placebo or probiotic capsules starting from the 16th week of gestation until birth. Each probiotic capsule contained >1 × 10^9^ colony forming units each of *Lactobacillus rhamnosus GG* and *Bifidobacterium lactis BB-12* per day. The primary outcome was diagnosis of GDM at 28 weeks of gestation, assessed by a 75 g oral glucose tolerance test [126]. Findings from the SPRING trial, involving 204 women in the placebo group and 207 women in the probiotics group (administered from the first half of the second trimester), concluded that the incidence of GDM was not reduced at the 28th gestation week for the probiotics group [126,127].

Lindsay and colleagues evaluated the use of *Lactobacillus salivarius UCC118* for a period of 4 weeks, starting from the 24th week of gestation, in a placebo-controlled, double blind, randomized trial, including 175 pregnant women with early pregnancy body mass index (kg/m^2^) between 30 and 39.9; 138 women completed the study (63 in the probiotic group and 75 in the placebo group), with no difference being found regarding maternal fasting glucose level, metabolic profile or pregnancy outcomes [122].

A systematic review and meta-analysis conducted by Jarde et al., counting 27 previous studies, found no evidence that administering probiotics or prebiotics during pregnancy influences the risk of preterm birth or the occurrence of GDM [128].

### 5.2. Treatment

Regular physical activity during pregnancy is beneficial in both healthy and GDM women. Pregnant women should be encouraged to initiate or continue exercise training during pregnancy, according to The American College of Obstetricians and Gynecologists (ACOG), providing additional benefits in the prevention of pregnancy-related complications [102]. The American Diabetes Association recommends exercise for a mean of 30 min almost daily for women with GDM, in order to ameliorate glucose control [2,3].

Women with GDM who frequently exercised during pregnancy (2–7 days per week, 30–60 min per session) proved to have beneficial outcomes not only concerning fasting blood glucose and postprandial blood glucose levels, but also fetal impact (lower birth weight) [129,130,131,132,133]. Some reports noted postponed insulin treatment [3,124,134].

Routine exercise during pregnancy proved to have favorable outcomes in GDM patients, such as better adjustments in skeletal muscles, ameliorated oxidative capacity, higher expression of proteins with roles in mitochondrial biogenesis, augmented lipid oxidation, and enhanced insulin sensitivity and glucose uptake. As a result, the inflammatory status can be diminished and the vascular function can be improved. Exercise provided benefits not only for the mother during pregnancy, but also for the postpartum period and fetal development [3].

It was reported that multispecies probiotic supplementation for a period of eight weeks in diabetic patients lowered inflammation, decreasing highly sensitive C-reactive protein serum levels [110,135]. On the contrary, there are some proper randomized controlled trials that point out the lack of influence of probiotics on fasting plasma glucose and fasting serum insulin levels, alongside gestational weight in GDM patients [110,136,137,138].

The systematic review and meta-analysis of six randomized controlled trials, involving 830 patients with GDM, pointed out that probiotic supplementation significantly reduced fasting serum insulin and insulin resistance, but there was no considerable impact on fasting plasma glucose, gestational age, and gestational weight [110].

According to the work of Zheng et al., who evaluated ten randomized controlled trials, probiotic supplements for women with GDM resulted in beneficial effects of glycemic control. However, the adequate dose and bacterial composition of probiotics and the long-term effects of probiotics used for pregnant women should be assessed further through large-scale clinical trials [138].

## 6. Conclusions

Gestational diabetes, its incidence and demographics, are changing worldwide, increasing attention on the subject. Up to date, there are conflicting data in the literature concerning the potential actions one can make in order to prevent or predict GDM in high-risk patients. The microbiota of women suffers a multitude of changes during pregnancy, mostly in the Firmicutes/Bacteroidetes ratio. Dysbiosis in early pregnancy, intertwined with host immunity, can influence later occurrence of GDM. Potential solutions for this problem can be lifestyle changes, such as regular physical exercise and diet changes, and even administering probiotics. Some studies suggest that probiotics can increase insulin sensitivity and diminish the inflammatory response, while also providing a better metabolic status. Unfortunately, there is no consensus regarding either the optimal dose and bacterial load of probiotics, or the adequate amount of time for the treatment. Further studies are needed in order to provide clear insights of GDM and its pathogenesis, alongside microbiota-targeted biomarkers for early diagnosis of GDM and potential prevention methods.

## Figures and Tables

**Figure 1 ijms-23-12839-f001:**
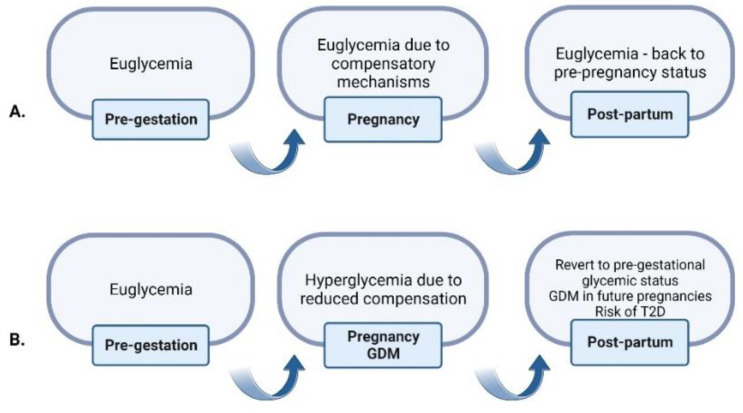
Glycemic status in normal pregnancy versus GDM. (**A**) In a normal pregnancy, the euglycemic status from the pre-gestational period will be preserved by the compensatory mechanisms (increasing blood glucose, hyperplasia and hypertrophy of β-cells, decreasing insulin sensitivity) which will return to normal after birth. (**B**) In a GDM pregnancy, these mechanisms fail to adapt, resulting in hyperglycemic status. After pregnancy, glycemic status can either return to normal or it can be a risk of T2D and of GDM in future pregnancy [17]. Created with BioRender.com (accessed on 8 October 2022).

**Figure 2 ijms-23-12839-f002:**
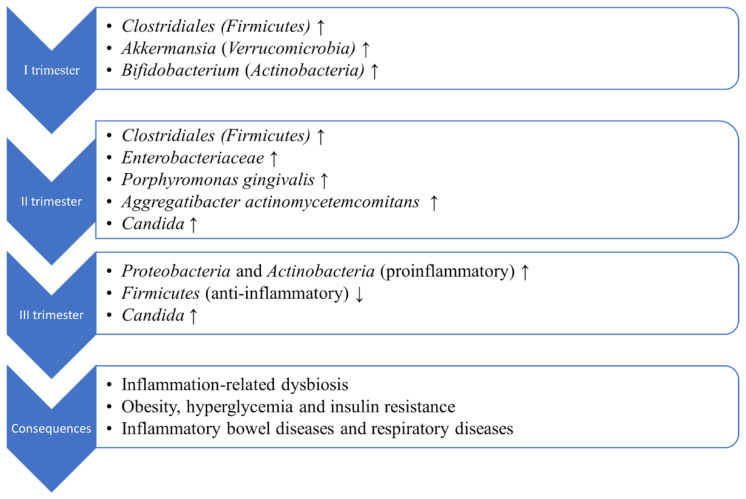
Evolution of the microbiota during pregnancy and its consequences.

**Table 1 ijms-23-12839-t001:** The most important modifiable and non-modifiable risk factors that can contribute to the development of GDM. One or more risk factors can be present in the same pregnant woman.

Risk Factors for GDM 2	
Modifiable	Non-Modifiable
Obesity [7,8,9]	Family history of GDM [8]
Change in weight between pregnancies [3]	Family history of T2D [8,10]
Significant weight gain in pregnancy [3]	Advanced age of the mother [11]
Stress [3]	Previous delivery of a baby >4000 g [8]
Antidepressant and psychotropic drugs [3]	High parity, hydramnios [3]
Smoking [3]	History of polycystic ovary syndrome [8]
Inadequate sleep patterns [12]	Ethnicity (African, Asian, American, Pacific Islands) [8,13]
Western diet [9]	Previous or pregnancy developed hypertension [12]
Sedentary lifestyle [8]	

**Table 2 ijms-23-12839-t002:** Dietary influence in gut microbiota composition.

Higher Fiber Diet	High Fat Diet	High Protein Diet	High Carbohydrate Diet
*Bacteroidetes* ↑ [60]	*Bacteroidetes* ↓ [61]	*Bacteroidetes* ↑ [62,63]	*Bacteroidetes* ↑ [64]
*Firmicutes* ↑ [65]	*Firmicutes* ↑/↓ [61,66]	*Firmicutes* ↑ [63]	*Firmicutes* ↑ [64]
*Actinobacteria* ↑ [65]	*Actinobacteria* ↓ [67]	*Proteobacteria* ↑ [63]	*Actinobacteria* ↑ [68]
*Proteobacteria* ↓ [67]	*Proteobacteria* ↓ [67]	*Deferribacteres* ↑ [63]

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
