# Peer review of "Gut Microbiome Changes in Gestational Diabetes"

_ijms, 2022, doi:10.3390/ijms232112839_

Round 1
Reviewer 1 Report
The manuscript "Gut microbiome changes in gestational diabetes" describe an overview of the correlation between gestational diabetes mellitus and dysbiosis and current and future methods for prevention and treatment (lifestyle changes, pre-and probiotics administration). The manuscript is generally written (in this form, the manuscript looks like an unfinished draft). The article's structure is ambiguous, a part of the ideas is repeated, and the analytical methods are not chronologically exposed. I suggest the authors identify the aim of the review.
1. I recommend rephrasing the abstract. The abstract should follow the style of structured abstracts but without headings: 1) Background: Place the question addressed in a broad context and highlight the purpose of the study; 2) Methods: Describe briefly the main methods or treatments applied. Include any relevant preregistration numbers and species and strains of any animals used. 3) Results: Summarize the article's main findings; and 4) Conclusion: Indicate the main conclusions or interpretations.
2. Introduction- Finally, briefly mention the main aim of the work and highlight the main conclusions.
3. I recommend checking the guide for citations in the text.
4. For Tables 1 and 2, please include more information: the clinical trials and the dose used for the test. Moreover, it is necessary to have also the experimental design of the problems and the description of the experiment.
5. Figure 2 The microorganism name should be in italic.
6. Line 152 I recommend that the authors use the same format for all the words except the microorganism name.
7. Lines 155-157 I recommend being specific by adding more details (%) "decreasing the Bacteroidetes phylum and increasing the Firmicutes phylum."
8. Lines 159-166; 176-183; 190-196; The information in the paragraph is very general. Please discuss this in more depth.
9. Table 2 - The microorganism name should be in italic
- I recommend using the same format font for the table. I recommend checking the guide for tables about the table format.
10. Line 184 Please specify a few examples for "numerous alterations."
11. Line 338 "several studies" I recommend specifying these studies.
12. Lines 342, 348, 350 ">1 x 109"; "28th" Please check the format.
13. Line 369 The microorganism name should be in italic.
14. References- Please check the format of the references. I recommend checking the guide for reference format.
Author Response
#Reviewer 1
The manuscript "Gut microbiome changes in gestational diabetes" describe an overview of the correlation between gestational diabetes mellitus and dysbiosis and current and future methods for prevention and treatment (lifestyle changes, pre-and probiotics administration). The manuscript is generally written (in this form, the manuscript looks like an unfinished draft). The article's structure is ambiguous, a part of the ideas is repeated, and the analytical methods are not chronologically exposed. I suggest the authors identify the aim of the review.
We acknowledge #Reviewer 1 for the comments that definitely helped us to improve the manuscript.
- I recommend rephrasing the abstract. The abstract should follow the style of structured abstracts but without headings: 1) Background: Place the question addressed in a broad context and highlight the purpose of the study; 2) Methods: Describe briefly the main methods or treatments applied. Include any relevant preregistration numbers and species and strains of any animals used. 3) Results: Summarize the article's main findings; and 4) Conclusion: Indicate the main conclusions or interpretations.
Thank you for this suggestion. We have restructured the abstract accordingly.
- Introduction- Finally, briefly mention the main aim of the work and highlight the main conclusions.
Thank you for this suggestion, we all considered it very useful. We added a paragraph including the requested information.
- I recommend checking the guide for citations in the text.
We did that. Thank you.
- For Tables 1 and 2, please include more information: the clinical trials and the dose used for the test. Moreover, it is necessary to have also the experimental design of the problems and the description of the experiment.
Thank you. We have included the references in the tables so that all readers can consult those studies.
- Figure 2 The microorganism name should be in italic.
We modified the Figure accordingly. Thank you.
- Line 152 I recommend that the authors use the same format for all the words except the microorganism name.
We corrected it.
- Lines 155-157 I recommend being specific by adding more details (%) "decreasing the Bacteroidetes phylum and increasing the Firmicutes phylum."
Thank you for the suggestion. We extended the paragraph with more information and %.
- Lines 159-166; 176-183; 190-196; The information in the paragraph is very general. Please discuss this in more depth.
Thank you for the suggestion, we included additional details.
- Table 2 - The microorganism name should be in italic
- I recommend using the same format font for the table. I recommend checking the guide for tables about the table format.
Thank you for this suggestion. We modified the table accordingly.
- Line 184 Please specify a few examples for "numerous alterations."
We modified it accordingly.
- Line 338 "several studies" I recommend specifying these studies.
We modified it accordingly.
- Lines 342, 348, 350 ">1 x 109"; "28th" Please check the format.
We modified it accordingly.
- Line 369 The microorganism name should be in italic.
We modified it accordingly.
- References- Please check the format of the references. I recommend checking the guide for reference format.
Thank you for this suggestion. We modified the format with MDPI style.
Reviewer 2 Report
My major concern is that such kind of review article should be submitted by the top-quality experts in the field, who are able to critically validate the available data, not simply reprint and combine them into new articles. I have not found any original papers published by neither the first nor co-authors related to the review topic.
I leave this decision to Editors.
Other minor comments:
References – please follow the journal guidelines.
Table 2, figure 2 – the names of bacteria phyla should be written in italic.
Table 2 – did the authors have a permission of Bruno Senghor et al. to modify their work?
Line 236 – please explain several acronyms like MyD88, RORγt, IFN-γ
Line 330 – please provide number of reference for Luoto Raakel.
Line 339 – please explain what is SPRING trial.
Author Response
Reviewer 2
My major concern is that such kind of review article should be submitted by the top-quality experts in the field, who are able to critically validate the available data, not simply reprint and combine them into new articles. I have not found any original papers published by neither the first nor co-authors related to the review topic.
I leave this decision to Editors.
We appreciate this, however except for this, all the comments for our paper were more of proof reading phase.
Other minor comments:
References – please follow the journal guidelines.
Table 2, figure 2 – the names of bacteria phyla should be written in italic.
Thank you. We corrected this.
Table 2 – did the authors have permission of Bruno Senghor et al. to modify their work?
We did not modify their work, we extracted partial information from a figure and collected it in a table. We performed additional collection of data and added the references in the table.
Line 236 – please explain several acronyms like MyD88, RORγt, IFN-γ
Thank you. We explained the acronyms.
Line 330 – please provide number of reference for Luoto Raakel.
Thank you. We introduced the reference.
Line 339 – please explain what is SPRING trial.
We have included more details about SPRING study and explained the acronym.
Round 2
Reviewer 1 Report
The authors have modified the signaled errors accordingly.
I consider that the paper was improved enough.